# Simultaneous Irradiation with UV-A, -B, and -C Lights Promotes Effective Decontamination of Planktonic and Sessile Bacteria: A Pilot Study

**DOI:** 10.3390/ijms241612951

**Published:** 2023-08-18

**Authors:** Andrea Bosso, Francesca Tortora, Rosanna Culurciello, Ilaria Di Nardo, Valeria Pistorio, Federica Carraturo, Andrea Colecchia, Rocco Di Girolamo, Valeria Cafaro, Eugenio Notomista, Raffaele Ingenito, Elio Pizzo

**Affiliations:** 1Department of Biology, University of Naples Federico II, 80126 Naples, Italy; francesca.tortora@unina.it (F.T.); rosanna.culurciello@unina.it (R.C.); ilaria.dinardo@unina.it (I.D.N.); federica.carraturo@unina.it (F.C.); vcafaro@unina.it (V.C.); notomist@unina.it (E.N.); 2Centre de Recherche Saint-Antoine (CRSA), Sorbonne University, Inserm, 75012 Paris, France; valeria.pistorio@inserm.fr; 3Hygiene Laboratory, Centro Servizi Metrologici e Tecnologici Avanzati (CeSMA), University of Naples Federico II, 80146 Naples, Italy; 4Physics Department “Ettore Pancini”, University of Naples Federico II, 80126 Naples, Italy; colecchiaandrea828@gmail.com; 5Department of Chemical Sciences, University of Naples Federico II, 80126 Naples, Italy; rocco.digirolamo@unina.it; 6Naturamla S.R.L., 00195 Rome, Italy; ingenito@naturamla.com; 7Centro Servizi Metrologici e Tecnologici Avanzati (CeSMA), University of Naples Federico II, 80126 Naples, Italy

**Keywords:** environmental disinfection, decontamination, ultra-violet device, biofilm, bacterial growth

## Abstract

Surfaces in highly anthropized environments are frequently contaminated by both harmless and pathogenic bacteria. Accidental contact between these contaminated surfaces and people could contribute to uncontrolled or even dangerous microbial diffusion. Among all possible solutions useful to achieve effective disinfection, ultraviolet irradiations (UV) emerge as one of the most “Green” technologies since they can inactivate microorganisms via the formation of DNA/RNA dimers, avoiding the environmental pollution associated with the use of chemical sanitizers. To date, mainly UV-C irradiation has been used for decontamination purposes, but in this study, we investigated the cytotoxic potential on contaminated surfaces of combined UV radiations spanning the UV-A, UV-B, and UV-C spectrums, obtained with an innovative UV lamp never conceived so far by analyzing its effect on a large panel of collection and environmental strains, further examining any possible adverse effects on eukaryotic cells. We found that this novel device shows a significant efficacy on different planktonic and sessile bacteria, and, in addition, it is compatible with eukaryotic skin cells for short exposure times. The collected data strongly suggest this new lamp as a useful device for fast and routine decontamination of different environments to ensure appropriate sterilization procedures.

## 1. Introduction

Contaminated surfaces, medical devices, surgical instruments, and even simple objects of common use play a significant role in the transmission of many pathogens [1]. This is the reason why effective procedures aimed at eliminating potential harmful microorganisms and viruses (disinfection techniques) are currently being widely investigated. Many of these techniques involve the use of chemicals like alcohols, chlorine compounds, aldehydes, hydrogen peroxide, iodophors, acids, quaternary ammonium compounds, and ozone [2,3]. While often quite effective, these methods present some limitations and drawbacks such as toxic by-products, possible pungent and suffocating odors, the corrosion of metals, incompatibility with rubber and plastic materials, and equipment costs (e.g., ozonation) [4,5]. Furthermore, the effectiveness of a given chemical disinfectant depends on its usage concentration, contact time, temperature, turbidity, shelf life/stability, pH, and specific microbial targets [6]. In addition, it should not be underestimated that some chemical agents must be accurately removed before being able to reuse the items subjected to disinfection. All these critical issues are widely recognized and represent the driving force behind the development of alternative disinfection methodologies not necessarily linked to the use of chemical agents [7,8,9]. Among the physical procedures able to significantly improve the sanitization process, UV light is currently attracting extensive attention as a validated technology for the disinfection of pathogens on surfaces, in the air, and in water, both in terms of safety and degree of effectiveness [10,11]. Ultraviolet (UV) radiation is a component of the electromagnetic spectrum that falls in the wavelength range of 100 to 400 nm and that can be further divided into four different categories, namely, UV-A (320–400 nm), UV-B (280–320 nm), UV-C (200–280 nm), and Vacuum-UV (100–200 nm). The UV-C spectrum has been employed extensively as the germicidal range of UV radiation [12]. Exposure to UV-C irradiation produces detrimental effects to bacteria and virus survival as a result of photochemical damage mainly at the expense of the pyrimidine bases of nucleic acids. The resulting formation of pyrimidine 6–4 pyrimidone and cyclobutane pyrimidine dimers (bipyrimidine photoproducts) leads to the inhibition of the correct transcription and replication processes [13,14]. However, it is noteworthy that UV-B (280–320 nm), in addition to UV-C, can exert direct harmful effects on nucleic acid bases [15], although nowadays their main use is not related to germicidal properties, but more aimed at phototherapy for the treatment of psoriasis and other skin diseases [16,17]. On the other hand, UV-A radiations (320–400 nm) seem to be responsible for indirectly killing bacterial cells by inducing reactive oxygen species (ROS) -mediated pathways [18] but, as with UV-B, they are mainly used for the treatment of dermatological diseases (such as atopic dermatitis, psoriasis, mycosis fungoides, alopecia areata, stretch marks, urticaria, etc.) [19]. Nonetheless, the germicidal effects of UV-A radiations on some bacteria [20,21] as well as the ability to affect bacterial biofilm by increasing the production of hydroxyl radicals such as 8-hydroxy-2′-deoxyguanosine have been reported [22]. All these statements gave us the inspiration to imagine a device capable of simultaneously exploiting the peculiar antimicrobial properties of the full UV spectrum. Conventional UV sources for disinfection systems include low- or medium-pressure mercury lamps able, respectively, to produce a single monochromatic wavelength of 254 nanometers or multiple types of wavelengths ranging from 200 to 400 nanometers [23]. Although still widely used, they present several critical issues as brittleness, toxicity (due to mercury), a relatively short lifetime, and a significant energy demand [24,25]. Nevertheless, in the past few years, with the rapid development and improvement of the semiconductor industry, UV light-emitting diodes (UV-LEDs) have emerged as a new source to generate UV radiation [22]. A growing number of convincing recent reports allow us to attribute the role of a promising alternative to conventional UV mercury lamps to UV-LEDs because, besides being environmental friendly (no mercury), they offer several unique advantages such as a great variety of wavelengths, compactness and robustness, faster start-up time, less energy consumption, longer lifetimes, and the ability to turn on and off with high frequency [26,27]. The scaffold of LEDs is based on the junction of two-terminal semiconductors, known as the p–n junction, which converts direct currents into radiation. Such devices can emit light at different wavelengths, but, above all, they have a specific characteristic to simultaneously emit light at various wavelengths at the same time [28]. This opens the possibility of combining different LED wavelengths, which could lead to a probable synergistic effect on bacterial inactivation and biofilm eradication [29,30,31]. Additionally, a treatment with a combination of different UV light wavelengths could enhance the microbial inactivation, attenuating more effectively damaged DNA repair mechanisms [32,33]. In recent years, the interest in understanding the potential efficacy of combined UV is increasing, but, to date, only a few reports, mostly limited to water disinfection studies, present significant evidence [34]. In order to preliminarily explore the potential of combined UV radiations including not only UV-C, but supported by properties of UV-A and UV-B [35,36], in this study we chose to investigate the effectiveness of a new-generation UV-LED lamp to cause microbial the inactivation of both planktonic and sessile bacterial strains. In this regard, we chose to test as a surface disinfectant device an innovative UV-LED lamp, not yet commercially available (see the Materials and Methods section) and conceived with a combination of three different UV light sources: UV-A (UV range: 315–400 nm; peak: 360 nm), UV-B (UV range: 280–325 nm; peak: 305 nm), and UV-C (UV range: 220–280 nm; peak: 260 nm). In this pilot study, the surface decontamination activity of combined UV radiations was analyzed on six bacterial strains and compared with that of a device emitting only UV-C radiations. Furthermore, the power to reduce bacterial spread and the property of eradicating bacterial biofilm of combined UV radiations were also analyzed. Finally, possible side effects of this new device on human keratinocytes have been analyzed and discussed.

## 2. Results

### 2.1. Combination of UV-A, UV-B, and UV-C Radiations: Determination of Irradiance and Energy Dose (Fluence)

All the experiments carried out in this work were conducted by positioning the UV-LED lamp (see the Materials and Methods section) at a height of 60 cm from the surface on which the biological samples were tested. Before proceeding with the biological tests, we measured the irradiance and energy dose (or Fluence) of the UV-LED lamp, considering the established operating height. These parameters allowed us to canonically determine the reliability of a UV device. The measurements were carried out at different exposure times (0–5–15–30–60–180 min) and the corresponding data are reported in Table 1.

### 2.2. Combined UV Radiations Possess Fast Antimicrobial Activity

As previously described, the underlying idea of this work was to verify the disinfectant power of a UV lamp capable of simultaneously emitting a combination of the three types of UV radiations (UV-A, UV-B, UV-C) (see Appendix A). To conduct this, we designed a first experiment using a series of individual bacterial plates on which to verify the action of the lamp positioned at a height of 60 cm.

Therefore, 1 × 10^7^ CFU of six different bacterial strains (*Staphylococcus aureus*, methicillin-resistant *Staphylococcus aureus*, *Enterococcus faecalis* as Gram-Positive and *Pseudomonas aeruginosa, Escherichia coli*, *Salmonella enterica* as Gram-Negative) were plated on Petri dishes and exposed for 30 s, 1 min, 2.5 min, and 5 min to the irradiation of the device placed at a height of 60 cm. As shown in Figure 1 (panels A and B), combined radiations exerted a strong antimicrobial effect already after 1 min and only *E. faecalis* required a longer exposure for the complete absence of colonies. In order to avoid any possible photo-transformation of the solid medium subjected to the UV lamp contributing to this decontamination effect, we also proceeded to verify if bacterial growth was affected on plates exposed for 5 min to UV radiation before bacterial seeding. As shown in Appendix A, solid media exposed to the UV lamp before bacterial seeding did not affect bacterial growth if compared to media exposed to UV radiation only after bacterial seeding, clearly indicating that the decontamination effects reported in Figure 1 were not linked to possible UV-promoted media transformation, but due to the interaction of UV light with contaminating bacteria.

The same conditions (initial CFU and time intervals) were also used in a control experiment, during which the lamp equipped with only UV-C was used (see irradiance, fluence, and technical data in Appendix A). As shown in the Appendix A, the susceptibility of the tested bacteria to a single UV-C radiation would appear to be less pronounced if compared with that found in the test in which combined UV radiations were used. This allowed us to clearly highlight that, under the chosen experimental conditions, the combination of the three types of UV radiations presented a microbicidal power significantly higher than that of UV-C radiation alone, with the latter canonically recognized as a germicide and widely used in decontamination procedures.

In light of such surprising data obtained with the collected bacteria, we wondered if this device was also able to act similarly on environmental bacterial strains isolated from different sources (see the Materials and Methods section), also considering the common discrepancy often observed when comparing responses of laboratory strains and strains isolated from the environment. Intriguingly, superimposable results were also highlighted by subjecting plates contaminated with environmental strains to the same disinfection procedure previously described. Indeed, as it is possible to observe in Figure 2A,B, the environmental bacterial strains were effectively eradicated already after 30 s of exposure to the UV lamp.

All tests conducted so far were carried out on solid culture media subjected to controllable contamination, conditions useful for reproducibility purposes. However, to verify the possible applicability of this innovative device in real conditions, it was decided to also analyze its effectiveness on alternative surfaces. The choice of surfaces to be taken into consideration fell on a polyvinyl chloride (PVC) panel and a ceramic tile to also highlight any possible differences between smooth (PVC) and rough (ceramic tile) surfaces. The surfaces were exposed to outdoor air for 24 h, allowing natural microbial contamination: microbiological tests were subsequently performed, sampling surfaces with contact plates before and after exposition to the UV lamp for 5, 60, and 180 min. The results show high death rates, between 83% and 94%, after just 5 min of exposure to combined UV radiations (Appendix A). Toxic effects corresponded to an almost maximum average death rate of bacteria, molds, and yeasts (97–100%) after 180 min, thus also confirming the previous results of the in vitro tests of antimicrobial activity (Figure 1 and Figure 2) and further indicating that there is no significant difference in the effects of the UV lamp between smooth and rough surfaces.

Finally, in order to understand if combined UV radiations could be compatible with eukaryotic cells, we investigated their effects on both metabolic activity and the release of ROS of human keratinocytes (HaCaTs) after up to 3 h of UV exposure. The metabolic activity of HaCaT cells exposed to combined UV radiations was measured via MTT assay, and it was observed that UV exposure slightly altered the viability of HaCaT cells only after 15 min, and serious toxicity was detected only after prolonged irradiation (up to 3 h) (Figure 3A). A similar trend was also obtained by analyzing the release of ROS in HaCaT cells subjected to combined UV radiations. In this case, the release of ROS was compared with that obtained when HaCaT cells were subjected to treatment with the oxidizing agent sodium arsenite (SA) [37,38]. As shown in Figure 3B, the combination of UV radiations resulted in an ROS release comparable to that induced by SA only after a 3 h exposure.

Both results shown in Figure 3A,B therefore suggest that the combination of the three UV radiations may cause serious consequences on eukaryotic cells only under prolonged exposure. As previously ascertained (Figure 1), the duration of the irradiation necessary for complete bacterial decontamination requires a rather narrow time frame in which we can reasonably assume that the potentially harmful effects of combined UV radiations on the human cells tested are absent or negligible.

In this regard, a further test useful to verify the response of the HaCaT cells subjected to the action of the three combined UV radiations was carried out. In this case, we monitored the possible formation of stress granules, membrane-less subcellular aggregates successfully used as markers of cells’ response to stress in several cell lines, including keratinocytes [39,40]. Therefore, in order to examine the possible formation of stress granules, HaCaT cells were exposed to the combined UV radiations for up to 15 min and, subsequently, their intracellular morphology was compared, both with SA-treated cells [41] and with unstressed cells, using confocal immunofluorescence. As shown in Figure 3C, we found that HaCaT cells subjected to combined UV radiations for 15 min were not prone to form stress granules. This result, further validated by the behavior of SA-treated HaCaT cells, allowed us to reasonably assume that UV-treated cells do not undergo a significant alteration in their homeostasis in the interval of time necessary to completely eradicate the bacteria from the irradiated surface. Overall, the evidence collected in the tests described in Figure 3 indicates that combined UV radiations have a strong immediate microbicidal power and, at the same time, negligible effects on eukaryotic cells.

### 2.3. Combined UV Radiations Reduce Bacterial Contamination and Delay Bacterial Growth

Humid environments such as plumbing pipes, sinks, toilets, as well as turbid environments, require more frequent disinfection than dry surfaces and tend to reduce the effectiveness of UV-mediated disinfection [42,43]. For this reason, we decided to investigate the ability of combined UV radiations to reduce bacterial contamination in a static liquid culture medium. For this purpose, 200 µL of 1 × 10^7^ CFU/mL of *Staphylococcus aureus*, methicillin-resistant *Staphylococcus aureus*, *Enterococcus faecalis, Pseudomonas aeruginosa, Escherichia coli*, and *Salmonella enterica* were seeded into a 96-well plate and exposed to combined UV radiations for up to 3 h. Subsequently, bacteria were subjected to serial dilutions, plated on solid medium, and, finally, after a further incubation at 37 °C for 16 h, CFUs were counted. As shown in Figure 4, it was found that combined UV radiations caused a time-dependent reduction in the number of CFUs of all analyzed bacterial strains, highlighting a marked effectiveness against *Salmonella enterica* and *Enterococcus faecalis*.

Moreover, we also evaluated if combined UV radiations were able to inhibit or slow active bacterial growth. To conduct this, 1 × 10^7^ CFU/mL of bacteria were plated on a 12-well plate; subjected to 30 min, 1 h, and 3 h of exposure with combined UV radiations; and, finally, placed in a 37 °C thermostatic plate reader under continuous stirring in which bacterial growth was monitored at a wavelength of 600 nm. Bacterial growth curves were recorded, and the time delay needed to achieve a mid-logarithmic phase between each UV-treated and untreated bacterium was determined and is reported in Figure 5A,B. In this case, it was observed that bacteria exposed to combined UV radiations had a growth delay tightly correlated to UV exposure time. These data would suggest that combined UV radiations could determine not only powerful bactericidal activity on bacteria that contaminate surfaces, but also a bacteriostatic effect on bacteria that are actively growing in the presence of a large amount of nutrients and oxygen.

### 2.4. Combination of UV-A, -B, and -C Radiations Effectively Hinders the Stability of Biofilm Matrix and Reduces the Viability of Biofilm-Embedded Bacteria

Biofilms are self-organized bacterial communities that are stable and difficult to eradicate and that can contaminate various infrastructural systems. The formation of such a stable matrix requires a long time; nevertheless, infrequent or inappropriate disinfection procedures can lead to the formation of biofilms. Planktonic bacterial cells are free to move, while biofilm-embedded bacteria tightly adhere to a surface and also establish cell-to-cell interactions through a complex extracellular matrix composed of polysaccharides, proteins, lipids, and extracellular DNA [44,45]. Such features confer a peculiar stability to the biofilms and significantly decrease the penetrability of bacteria to antimicrobial agents. Once the germicidal power of combined UV radiations on planktonic bacteria was established, we proceeded to verify whether this UV device possessed the same power on a preformed biofilm. To conduct this, the same above-mentioned bacterial strains were grown in Mueller–Hinton Broth (MHB) for 48 h at 37 °C to promote the formation of stable biofilms and subsequently subjected to treatment with combined UV radiations for 1 or 3 h. The percentage of biofilm mass and metabolic activity of the bacteria trapped inside the biofilm were then measured (see the Materials and Methods section). The results reported in Figure 6A,B show clear time-dependent effects induced by combined UV radiations both on bacterial cell viability and on biofilm biomass. These results seemed quite interesting, as it can be assumed that the combined UV radiations possess additional properties not only limited to direct cytotoxicity, but which are also useful for disintegrating complex aggregates such as those present in bacterial biofilms.

In this regard, a further test was performed using Scanning Electron Microscopy (SEM) with the aim of better-understanding the effects of combined UV radiations on biofilm morphology. In this case, we selected two different bacterial strains: *Salmonella enterica* 706 RIVM and *Staphylococcus aureus* ATCC 12600. Both bacterial strains were seeded on slides, grown in MHB for 48 h at 37 °C, and then exposed to combined UV radiations for 1 or 3 h. The bacterial specimens were then fixed with glutaraldehyde, dehydrated with increasing concentrations of ethanol, and metallized to observe the morphology using SEM.

As shown in Figure 6C, already after 1 h of exposure to combined UV radiations, the normal 3D morphology of a stable biofilm generated by both a Gram-positive bacterium (*S. aureus*) and a Gram-negative bacterium (*S. enterica*) was highly compromised (gathered bacteria). Moreover, when the UV exposure time was extended to 3 h, biofilms lost connection between the cells and the original structure of the biofilm was further damaged. In addition, a significant reduction in the number of bacteria attached to the surface was also highlighted. Overall, the data collected with SEM fully agree with the data obtained by measuring the total biomass with the crystal violet assay or cell viability via MTT, highlighting the capability of combined UV to interfere not only with the viability of bacteria embedded into the biofilm, but also with biofilm’s structural integrity and compactness, thus favoring an effective disinfection procedure.

## 3. Discussion

Shared and crowded environments are a potential source of pathogen transmission and are rarely subjected to rigorous and effective cleaning procedures [4,46]. This is even more decisive if we consider that we are in a historical period plagued by new viral pandemics and by the consolidated spread of the phenomenon of bacterial antimicrobial resistance (AMR) [47,48]. In this scenario, several non-contact disinfectant methods are currently under development and, among them, UV irradiation clearly presents the most beneficial applicability [49] due to its wide-spectrum activity in addition to avoiding issues linked to possible side effects often produced by the use of chemical sanitizers [50]. The inactivation of microorganisms using UV-C light-induced damage to DNA and RNA is widely known and, for this reason, often at the basis of disinfectant UV devices [13]. However, what emerges from recent studies is that also UV-B and UV-A have a significant microbicidal power, and this assumes particular importance if we consider that their targets are different [51]. All this led to the conception of an attractive hypothesis based on the idea of being able to use the three different radiations simultaneously and exploit their complementarity to hinder the possible defenses of microorganisms to the extreme, thus maximizing the microbicidal effect. To preliminarily verify this, we probed the potential of the three combined radiations by imagining applying the action of the UV device to a hypothetical surface contaminated by bacteria. The idea also arose from the fact that, to the best of our knowledge, to date, the combined action of the three types of UV radiation has been investigated only in water disinfection [52] and never investigated for surface decontamination.

Based on the foregoing information, in the current study we tested the antimicrobial potential of combined UV radiations using an innovative LED device equipped with three sources able to simultaneously emit UV-A, UV-B, and UV-C radiations. The calibrated control of the simultaneous irradiations and the numerous advantages that LED technology offers if compared with the canonical mercury UV lamps (e.g., lower energy requirements, long lifespan, minimal warm-up time, greater robustness, etc.) [27] were essential for our purposes. All tests were carried out by positioning the UV-LED device at a height of 60 cm from the surface on which the various cell cultures rested. In this way, we were thus able to ascertain that the combined action of the three radiations effectively expressed a decidedly more marked microbicidal activity than the action induced by UV-C radiation alone. Indeed, simulated contaminated surfaces composed of both conventional laboratory bacterial strains and environmental isolates were eradicated from Petri dishes with an incredibly short exposure time (often less than 1 min) of combined UV lights. Conversely, UV-C radiation alone did not guarantee similar results for the same strains up to 5 min. These results clearly indicate that this new combination is particularly effective in eliminating bacteria living on a solid substrate. In the Appendix A, we also described the same effect reproduced on two other alternative contaminated surfaces, a smooth panel and a rough tile: in both cases, very promising results were obtained after just 5 min, reducing contamination by 93% of total bacteria and 85% of molds and yeasts.

We also wondered whether the bacterial life in water and low nutrient availability could reduce the effectiveness of the antimicrobial properties of combined UV. Our data, however, indicate that almost all bacteria were abundantly eliminated after only 30 min. Only *Escherichia coli* and *Salmonella enterica* required a prolonged time of exposure; however, this did not exceed 3 h. Equally interesting was also the effect of the UV-LED device exerted on actively growing bacteria with a large availability of nutrients. In this case, bacterial cells exposed to combined UV radiations presented a significant growth delay, thus highlighting that combined UV radiations also show bacteriostatic properties likely associated with their different targets, which include, beyond gene nucleotides, also those belonging to key coenzymes of metabolism (e.g., NAD and FAD) which finally lead to excessive and harmful release of ROS [53,54].

The promising antimicrobial power of the combined use of the three UV radiations was finally detected also on sessile bacteria. In this case, we were able to ascertain that the device allowed us to drastically alter the three-dimensional morphology of the biofilm, whether generated by Gram-negative or -positive bacteria, but also to effectively interfere with the vitality of the bacteria themselves. This further strengthens the hypothesis that the combined use of the three radiations makes it possible to hit diversified targets and, therefore, to carry out the disinfectant action more effectively. This idea consequently leads us to imagine that the intrinsic synergy between the three types of radiation could allow more targeted and shorter use of UV-LED lamps in widely anthropized contexts (e.g., offices, laboratories, hospitals, etc.), thus limiting the spread of potentially infectious microorganisms, also with the guarantee that, as evidenced by the data collected in Figure 2, any accidental exposure of human skin cells to this lamp does not lead to adverse events in terms of vitality and cellular stress. The findings deriving from this study imply that the development of new technologies suitable for the disinfection of objects or heterogeneous environments, whether ordinary or extraordinary, can reasonably involve also the use of combined UV radiations as an effective solution to reduce the propagation of pathogens.

## 4. Materials and Methods

### 4.1. Irradiance and Energy Dose Measurement

The UV device equipped with UV-A, -B, -C channels (technical data reported in Appendix A—Appendix A) was placed at a distance of 60 cm from samples in all the following experiments and its irradiance and energy dose values were measured using an IRRADIANCE probe LP 471 with SICRAM module (purchased from Delta OHM S.r.l., Padova, Italia). Irradiance (E) is the density of radiation incident on a given surface. If the irradiance is continuous and constant in time, it can be defined as the Fluence (F): F is given by E × t, where t is the exposure time in seconds. It is common to use mW/cm^2^ for fluence rate or irradiance and mJ/cm^2^ for fluence (UV dose) [55,56].

### 4.2. Bacterial Strains

All bacteria used for antibacterial and anti-biofilm experiments [57] are listed below: *MRSA* WKZ-2, *Salmonella enterica* 706 RIMV, *Pseudomonas aeruginosa* O1, *Escherichia coli* ATCC 25922, *Enterococcus faecalis* ATCC 29212, and *Staphylococcus aureus* ATCC 12600 (ATCC^®^, Manassas, VA, USA). The following environmental bacterial strains were also used: *Escherichia coli* strain LP50 and *Pseudomonas aeruginosa* strain Sihong_639 1 (isolated from water from a civil treatment plant), *Enterococcus faecalis* strain 2675, *Staphylococcus aureus* strain MS1-4 and methicillin-resistant *Staphylococcus aureus* strain TPS3156 (isolated from a milk processing plant), and *Salmonella enterica* subsp. enterica serovar Typhimurium strain LT795114.1 (isolated from an untreated drinking water spring).

### 4.3. Antimicrobial Activity

Bacteria were grown at 37 °C in Luria–Bertani (LB) broth until reaching the logarithmic phase, then each strain was diluted to 1 × 10^7^ CFU/mL and was seeded on solid culture medium in Petri dishes. Plates were exposed to both the combined UV radiation device and the selected UVC lamp for 5, 2.5, 1, and 0.5 min and incubated over night at 37 °C. The following day, the differential bacterial growth on plates exposed to UV irradiation and negative control (unexposed plates) was evaluated [58,59]. The plate coverage, expressed as percentage, was calculated on threshold- and intensity-inverted regions of plates by using ImageJ Version 1.53t bundled with Java 8 (https://imagej.nih.gov, accessed on February 2023; U. S. National Institutes of Health, Bethesda, MD, USA).

In order to simulate a possible decontamination in liquid medium, bacteria were diluted to 1 × 10^7^ CFU/mL in 200 µL of 10% Nutrient Broth (NB) in PBS and treated with the UV device up to 180 min. Then, bacteria were subjected to serial dilutions, seeded on LB agar plates, grown for 16 h at 37 °C, and then the bacteria colonies were counted [60,61,62].

To determine the bacterial growth slowdown after treatment with the lamp, bacteria were seeded in 12-well plates and exposed to the UV lamp for 30, 60, and 180 min. After treatments, bacterial growth was followed by measuring OD_600_ every 30 min. Bacterial growth delay was defined as the delay in hours for UV-treated bacteria to reach half of the maximum OD of the untreated control (1/2 OD_600_) [63,64].

To verify the microbicidal activity of the UV-LED lamp, an on-field hygiene monitoring of specific safety and quality parameters, before and after the application of sanitization protocols, was performed on two naturally contaminated commonly employed surfaces: a polyvinyl chloride (PVC) panel and a rough ceramic tile.

The surfaces were exposed to outdoor air for 24 h, allowing natural microbial contamination. Microbiological tests were subsequently performed, sampling surfaces with contact plates according to ISO 18593:2018 (International Standards Organization, 2018) before and after the 5, 60, and 180 min of UV exposure. Microbiological analyses were conducted following the specific ISO guidelines, targeting total bacterial counts at 22 °C and 37 °C (using Plate Count Agar contact plates, Thermo Scientific, ISO 4833:2013) (International Standards Organization, 2013), and total mold and yeast countss (Rose Bengal Agar with chloramphenicol, Thermo Scientific, ISO 21527:2008) (International Standards Organization, 2008). Results are expressed in CFU/100 cm^2^.

### 4.4. Antibiofilm Activity

Bacteria were grown overnight in LB and then diluted to 1 × 10^8^ CFU/mL in MHB in 96-well plates. Biofilms were formed for 48 h at 37 °C and then exposed to the UV lamp for 30 min, 1, and 3 h. The percentage of total biomass was evaluated using crystal violet coloration as in [57,65] and the bacterial viability was assayed with MTT, as already reported in [63,66].

### 4.5. Scanning Electron Microscopy (SEM)

Scanning Electron Microscopy (SEM) analyses were performed on preformed biofilms of *S. enterica* 706RIM and *S. aureus* ATCC 12600 exposed for 60 and 180 min to the UV-LED lamp. After treatments, samples were processed and characterized as previously reported [63,67].

### 4.6. Metabolic Activity of Eukaryotic Cells

Immortalized human keratinocytes, HaCaTs, (5 × 10^3^ cells) were seeded into a 96-well plate and incubated for 24 h at 37 °C in presence of 5% CO_2_. Then, cells were exposed to UV irradiation for up to 180 min. Hence, the MTT assay was performed as already reported in [39,66]. Cell viability was expressed as the standard error mean percentage compared with unexposed cells used as control.

### 4.7. ROS Quantification Assay

The ROS quantification assay was carried out as already reported in [39] by using DCFH-DA (2′,7′-dichlorofluorescin diacetate), a non-fluorescent probe that, after diffusion into the cell, is deacetylated by the cellular esterases and then oxidized in presence of ROS into the fluorescent product 2′,7′—dichlorofluorescein (DCF). Briefly, 2 × 10^4^ cells were seeded into 96-well plates and incubated at 37 °C in a 5% CO_2_ atmosphere for 24 h. Then, after removing the medium, the cells were washed with PBS 1× buffer and exposed up to 180 min to the UV lamp. After the treatments, cells were incubated with 20 µM DCFH-DA diluted in PBS 1× at 37 °C for 40 min. After the incubation time, the fluorescence of the cells from each well was measured and recorded in the excitation/emission wavelengths of 485–532 nm, by means of a Multi-Mode Microplate Reader (Synergy™ H4). Results were then expressed as standard error mean of the fluorescence intensity, expressed as arbitrary units, compared with unexposed cells used as control.

### 4.8. Immunofluorescence

Immunofluorescence analyses were performed as already reported in [39]. Briefly, 2.5 × 10^4^ HaCaT cells were seeded on coverslips, cultured for 24 h at 37 °C and 5% CO_2_, and then exposed for increasing times to UV irradiation. A total of 500 µM sodium arsenite (SA) was used as positive control of oxidative stress and stress granule recruitment in HaCaT cells. Coverslips were mounted in Mowiol^®^ 4-88 and images were acquired using Zeiss Confocal Microscope LSM 700 at 63× magnification.

### 4.9. Statistical Analyses

Statistical analyses were carried out using GraphPad Prism Version 9.5.1 (Graphstats Technologies, Bangalore, India). Values are reported as the means ± SD of biological replicates (* *p* < 0.05, ** *p* < 0.01, *** *p* < 0.001, and **** *p* < 0.0001) compared with the respective controls (one-way ANOVA, followed by Bonferroni’s post-test).

## 5. Conclusions

Contaminated surfaces can harbor and transmit pathogens in different environments and are challenging to disinfect, but UV irradiations are currently emerging as one of the most “Green” technologies useful to achieve effective disinfection. Here, we investigated the antimicrobial and antibiofilm properties of an innovative UV device equipped with LED technology, which avoids the use of mercury and can emit ultraviolet light in a broader wavelength spectrum compared with the most commonly used UV lamps currently available. Indeed, the here-described lamp can emit light ranging from UV-A, -B, to -C, generating an overall disinfectant effect due to the simultaneous irradiation of the entire spectrum of UV light. We reported that this innovative device can quickly promote surface decontamination from planktonic or adhered bacteria and is effective against more persistent contamination like actively growing and biofilm-forming bacteria. Moreover, using human skin cells as a model, we have also highlighted that any accidental and limited exposure to this UV light source does not cause significant adverse side effects. In conclusion, we believe that this preliminary study, even though it is limited to bacteria and only one distance away from the contaminated surface, can stimulate the development of so-conceived devices and the application of such UV lamps in various widely anthropized contexts such as offices or hospitals, thus limiting the spread of infections.

## Figures and Tables

**Figure 1 ijms-24-12951-f001:**
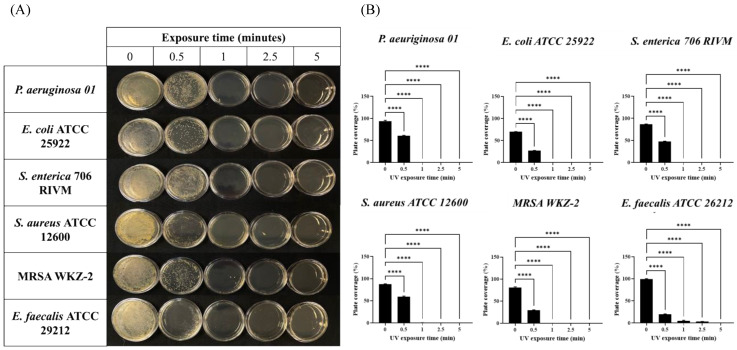
Effects of combined UV radiations on six different bacterial strains plated on Petri dishes. (**A**) Plates seeded with collection bacterial strains exposed to UV radiations for 0.5, 1, 2.5, and 5 min; (**B**) analysis of plate coverage (expressed as percentage) after UV exposure of collected bacterial strains for 0.5, 1, 2.5, and 5 min. Values are reported as the means ± SD of biological replicates (**** *p* < 0.0001) compared with the respective controls (one-way ANOVA, followed by Bonferroni’s post-test).

**Figure 2 ijms-24-12951-f002:**
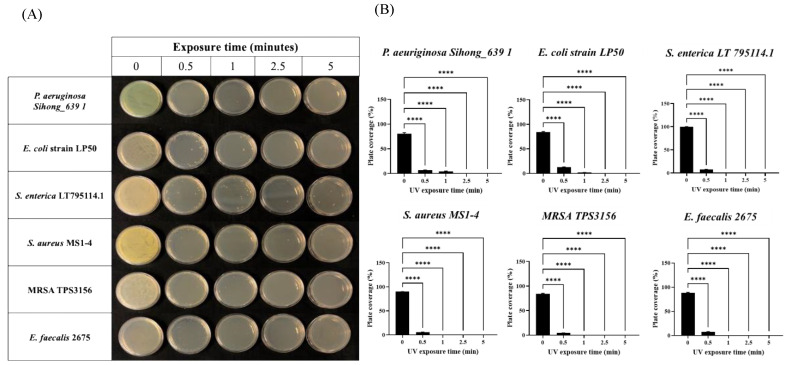
Effects of combined UV radiations on six different environmental bacterial strains plated on Petri dishes. (**A**) Plates seeded with environmental bacterial strains exposed to UV radiations for 0.5, 1, 2.5, and 5 min; (**B**) analysis of plate coverage (expressed as percentage) after UV exposure of environmental bacterial strains for 0.5, 1, 2.5, and 5 min. Values are reported as the means ± SD of biological replicates (**** *p* < 0.0001) compared with the respective controls (one-way ANOVA, followed by Bonferroni’s post-test).

**Figure 3 ijms-24-12951-f003:**
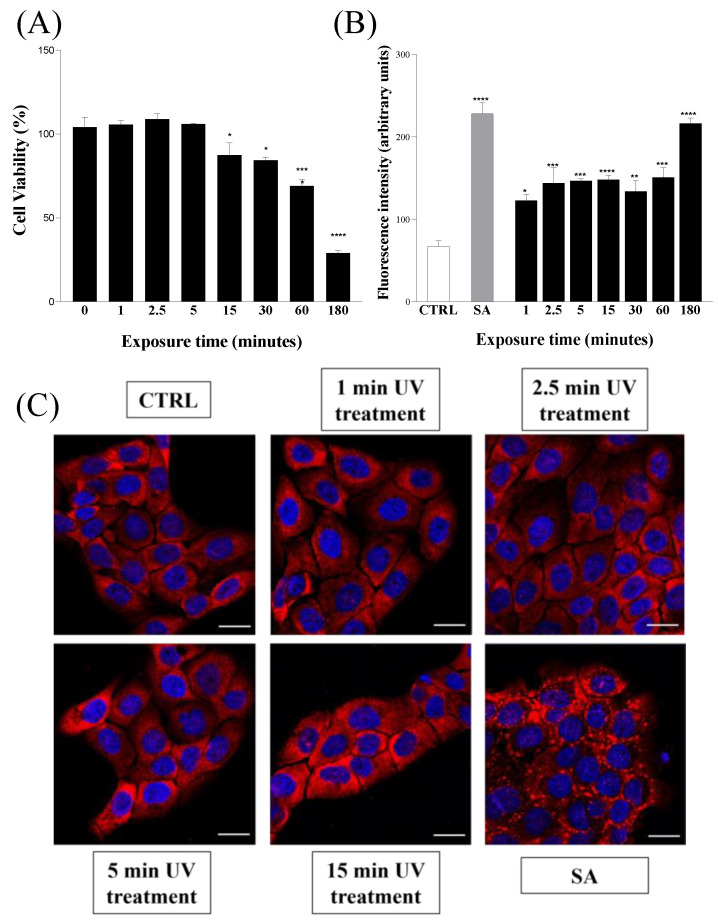
(**A**) Cell viability assay via MTT. HaCaT cells were exposed to combined UV radiations for seven different time intervals from 0 to 180 min. (**B**) Reactive oxygen species (ROS) detection via DCFH-Da assay. HaCaT cells were exposed to combined UV radiations for different time intervals ranging from 0 to 180 min. (SA) HaCaT cells treated with 500 µM sodium arsenite. (**C**) Stress granule detection via immunofluorescence analysis. HaCaT cells were exposed to combined UV radiations for four different time intervals ranging from 0 to 15 min. (CTRL) untreated cells; (SA) HaCaT cells treated with 500 µM sodium arsenite. Scale bars correspond to 20 μm in all cases. Values are reported as the means ± SD of biological replicates (* *p* < 0.05, ** *p* < 0.01, *** *p* < 0.001, and **** *p* < 0.0001) compared with the respective controls (one-way ANOVA, followed by Bonferroni’s post-test).

**Figure 4 ijms-24-12951-f004:**
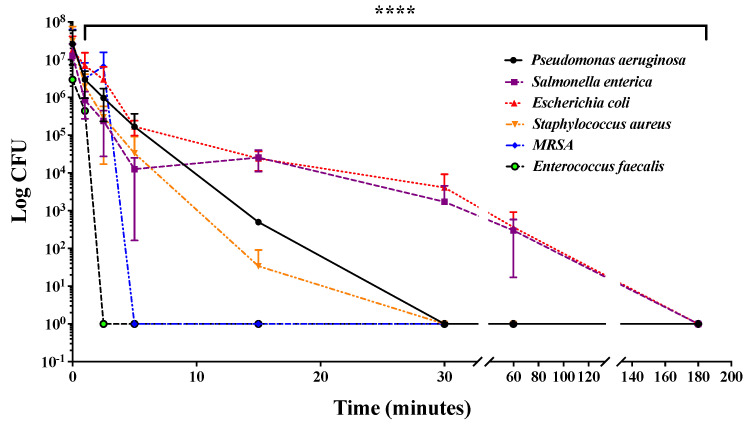
Time-dependent effects of combined UV radiations on the bacterial load of six different strains. CFUs corresponding to *P. aeruginosa*, *S. enterica*, *E. coli*, *E. faecalis*, *S. aureus*, and *MRSA* were determined after treatment with UV irradiation up to 3 h each compared with its negative control. Values are reported as the means ± SD of biological replicates (**** *p* < 0.0001) compared with the respective controls (one-way ANOVA, followed by Bonferroni’s post-test).

**Figure 5 ijms-24-12951-f005:**
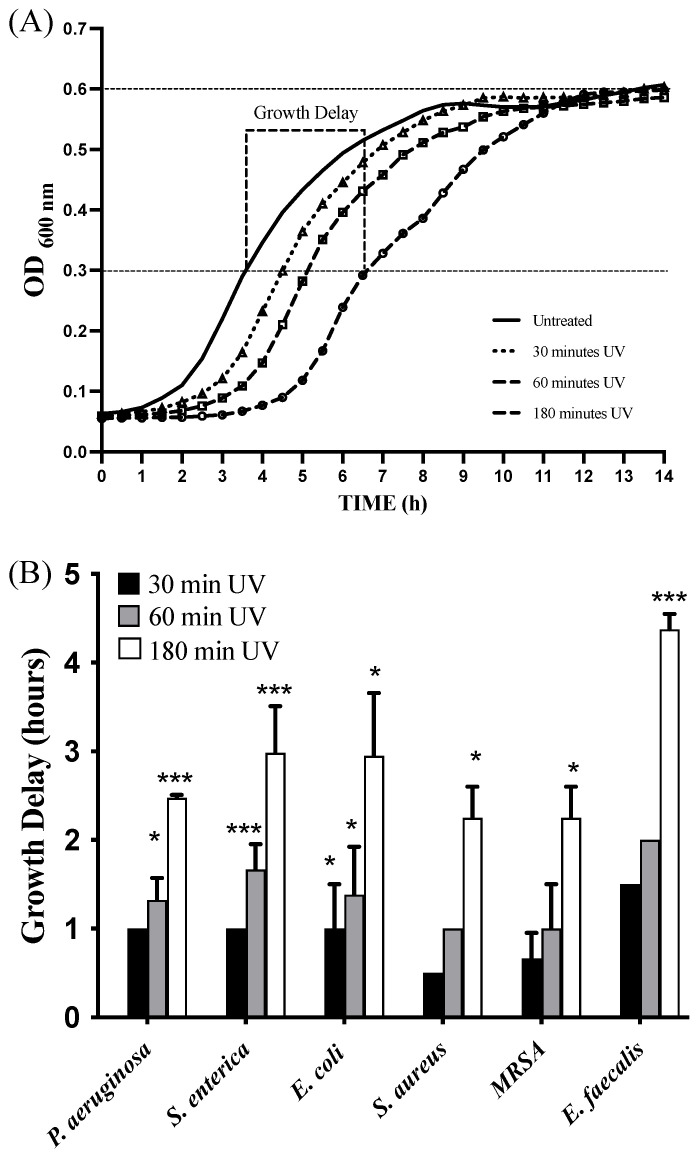
Bacterial growth delay of combined UV radiations. Bacterial strains were first exposed to combined UV radiations for three different time intervals (30, 60, and 180 min). (**A**) Subsequently, the growth of each UV-treated strain was monitored by measuring OD_600_. Growth delay was defined as the time needed for UV-exposed bacteria to grow above an ½ OD_max_ of control bacteria. (**B**) Time required for each bacterial strain to reach ½ OD_max_ following treatment with the UV lamp for 30–60–180 min with respect to the control. Values are reported as the means ± SD of biological replicates (* *p* < 0.05 and *** *p* < 0.001) compared with the respective controls (one-way ANOVA, followed by Bonferroni’s post-test).

**Figure 6 ijms-24-12951-f006:**
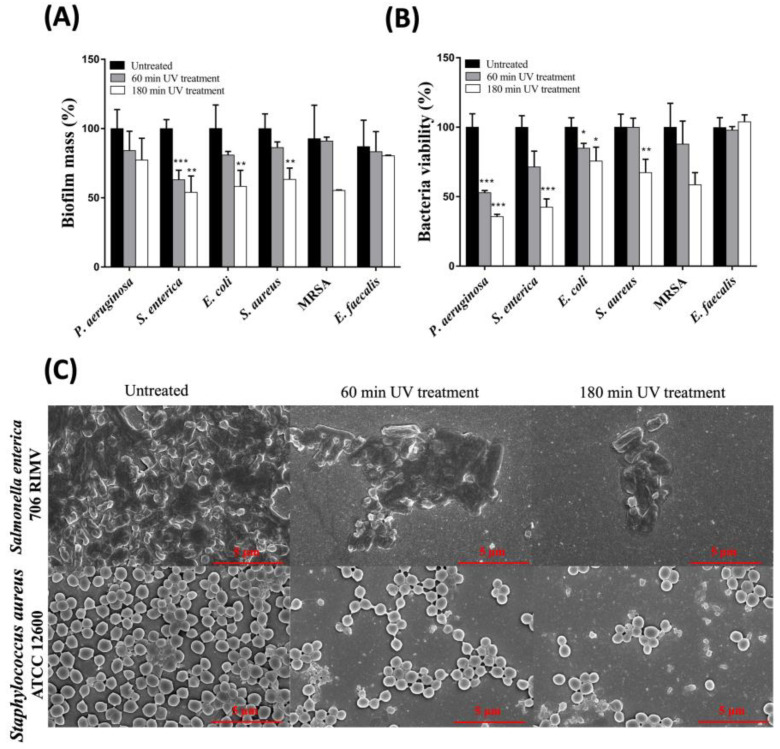
(**A**) Biofilm eradication exerted by combined UV radiations. Antibiofilm activity at different exposure times was tested on bacterial strains using crystal violet assay. (**B**) Cytotoxic effect of combined UV radiations on bacteria strains. The determination of bacteria viability within the biofilm after treatment with combined UV radiations was carried out using MTT assay. (**C**) Imaging at SEM. Biofilms of *S. enterica* 706RIM and *S. aureus* ATCC 12600 exposed at combined UV radiations for 1 and 3 h were subjected at imaging with SEM (magnitude 25,000×). Values are reported as the means ± SD of biological replicates (* *p* < 0.05, ** *p* < 0.01, and *** *p* < 0.001) compared with the respective controls (one-way ANOVA, followed by Bonferroni’s post-test).

**Table 1 ijms-24-12951-t001:** Irradiance and energy dose measured with all UV channels turned on simultaneously.

	UV-C (2 × 120 W)		UV-B (60 W)		UV-A (60 W)	
Time (min)	Irradiance (mW/cm^2^)	Energy Dose (mJ/cm^2^)	Irradiance (mW/cm^2^)	Energy Dose (mJ/cm^2^)	Irradiance (mW/cm^2^)	Energy Dose (mJ/cm^2^)
0	0.93	27.9	0.462	13.86	0.4	12
5	0.34	102	0.342	102.6	0.322	96.6
15	0.266	239.4	0.328	295.2	0.205	184.5
30	0.247	444.6	0.318	572.4	0.202	363.6
60	0.23	828	0.305	1098	0.19	684
180	0.21	2268	0.285	3078	0.175	1890

## Data Availability

All data generated or analyzed during this study are included in this published article (and its Appendix A).

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
