# Peer review of "Simultaneous Irradiation with UV-A, -B, and -C Lights Promotes Effective Decontamination of Planktonic and Sessile Bacteria: A Pilot Study"

_ijms, 2023, doi:10.3390/ijms241612951_

Round 1

Reviewer 1 Report

The article entitled "Simultaneous irradiation with UV-A, -B and-C lights promotes effective surface decontamination of both planktonic and sessile bacteira" reports an experimental study on the effect of a new UV device emitting light at different wavelenght for the decontamination of surfaces. The subject is in accordance with the topic of the journal in whish it could be piblished. It is well-written. To my point of view, knowing that I'm not en native English speaker, the English language if fine, easily understandable.

The methodology is deeply presented in the materials and methods part of the article and well-adapted for the goals fixed by the authors. The results are clearly presented and the conclusions are enough supported by the performed experiments, tables and figures shown, especially thanks to the additionnal files.

Howerver, I still have few questions to improve my understanding of this work.

i) Can the authors explain how they have chosen the exposure times used in the different experiments ?

ii) In line 143, the authors indicate that for MRSA the time for complete absence of colonies is longer than for the other bacteria strains. According to Figure 1, I'm not sure to have te same conclusion. To my point of view, it seems that it is for E. Faecalis that the needed UV exposure time is longer. Can the authors explain why they gave this conclusion ?

iii) For results presented from line 188 to 236, is it possible for the authors to give the environmental conditions in which the experiments were carried out (i.e. temprature, pressure, relative humidity, etc.) ?

iv) It can be obesrved that, especially for E. faecalis, the experiments did not give the same tendency due to different conditions (for instance results describes in Figures 1, 3, 5). How the authors can explain the obesrved differences for a same bacteria strain ?

v) The authors compared the effects of combined UV wavelenghts with UV C only. Would it have been also possible, in order to further explain the obtaines results, to compare with UV A alone, UV B alone ans combinations 2 by 2 ?

Finally, the discussion part (from line 343 ti line 397) appears more as a mixture of the introduction and conclusion than a real discussion of the results, which are very descriptive. The reader would like to learn more on how the different UV wavelenghts act on the different tested bacteria strains, why some bacteria strains present different results (is it due to the wavelenght answer ?), etc....

Author Response

Please see the attachment. Author responses are highlighted in red.

Reviewer 2 Report

This manuscript reports findings of a study on irradiation with UV light that promotes effective surface decontamination of bacteria. Although the overall topic might be of interest for scholars related to the field of desinfection and hygiene, there are some issues that limit the merit of this article. First of all, the text is not structured according to MDPI guidelines regarding the order of the main headings. Also, the text should be checked for grammar and style issues.

Please think of a more concise title and remove the .

Some of the keywords are not related to the content of the article.

The introduction lacks a clear rationale behind the study. Please remove the reference to the suppl. material from the introduction.

In the very last part of the conclusion section, we learn that this is a preliminary study. This fact should be elaborated on from the beginning and maybe even in the title so that readers are informed on this aspect earlier.

Practical and theoretical implications of the study findings should be clearer.

The text should be checked for grammar and style issues

Author Response

(The authors gave the same response as above.)

Round 2

Reviewer 2 Report

I thank the authors for providing a revised version of their article. I think that it should already be indicated in the title, that this is a pilot study.
The tables are pictures, not word tables.

(SUGGESTION: minor revision. )

The manuscript should be checked for grammar and style issues.

Round 3

Reviewer 2 Report

I thank the authors for modifying their manuscript. I recommend enhancing readability by using paragraphs to structure the text.

 Moderate editing of English language and style required